# OpenReview forum: "Concept-Based Dictionary Learning for Inference-Time Safety in Vision–Language–Action Models"
_ICLR.cc/2026/Conference — ICLR 2026 Conference Withdrawn Submission_

### Official Review · Reviewer_ieN5 · 2025-10-30

**Soundness:** 3
**Presentation:** 2
**Contribution:** 4
**Rating:** 4
**Confidence:** 4

**Summary:**

This paper introduces a novel, concept-based, inference-time safety framework to address the safety challenges of Vision-Language-Action (VLA) models in physical environments. Moving beyond conventional defenses at the input or output layers, this method delves into the model's hidden activation layers. It operates by learning a sparse and interpretable "concept dictionary" composed of both "safe" and "unsafe" concepts. During inference, this framework decomposes the model's internal state in real-time. When the activation strength of a harmful concept exceeds a predefined threshold, the system actively suppresses these activations, thereby "correcting" the model's behavioral intent without halting the task. This allows it to effectively defend against both explicit malicious instructions and adversarial jailbreak attacks.

**Strengths:**

- The paper's most significant contribution is its focus on defending within the model's latent representation space. This inference-based approach is quite refreshing. The fact that it's a plug-and-play module that obviates the need for retraining large VLA models is a huge advantage for real-world deployment, offering significant cost and efficiency benefits.

- The proposed activation attenuation strategy is particularly elegant. Instead of crudely halting the model, it gently weakens the intensity of harmful intents, guiding the model back onto a safe behavioral trajectory. The experimental results effectively demonstrate the method's ability to strike a fine balance between safety and utility.

**Weaknesses:**

- I don't have major reservations about the paper's technical contributions. However, I have to point out that the visual presentation quality is subpar and appears rushed. In Figures 1, 2, and 3, the choice of fonts and borders is quite unprofessional. Figure 2 is particularly problematic; its confusing layout makes it very difficult to intuitively grasp the framework. This lack of polish not only detracts from the paper's professionalism but also creates a significant barrier to understanding. I strongly recommend that the authors undertake a major revision of these figures to improve their clarity, logic, and aesthetic quality. Frankly, unless these presentation issues are thoroughly addressed, I would be hesitant to recommend this paper for acceptance.

- The Static Nature and Scalability of the Concept Dictionary: The framework's effectiveness is highly dependent on the pre-constructed concept dictionary. This implies it can only defend against harmful concepts that are already known and included in the dictionary, which is a core limitation of the method. The authors should provide a more in-depth discussion of this limitation and propose potential strategies to address it, such as methods for dynamically updating or expanding the dictionary to adapt to open-world environments.

**Questions:**

N/A

---

### Official Review · Reviewer_qdbp · 2025-10-31

**Soundness:** 2
**Presentation:** 2
**Contribution:** 2
**Rating:** 2
**Confidence:** 4

**Summary:**

The paper proposes an inference-time, concept-dictionary-based safety guard for VLA models. The method is evaluated on a safety-augmented Libero variant (“Libero-Harm”) and several recent VLA/robotic jailbreak suites, where it reportedly reduces attack success while preserving task success.

**Strengths:**

1. The paper is easy to read.

2. The approach is plug-and-play and does not require retraining the underlying VLA, which is practical for robotic tasks.

3. The experimental section covers several public safety/jailbreak suites and includes ablations over the gating hyperparameters, suggesting the technique is at least internally stable on their setup.

**Weaknesses:**

**1. Overstated problem novelty and framing.**

The paper claims to be “the first to formally define and unify the VLA Safety Problem.” However, safety in embodied/robotic agents, including settings where instructions can be malicious or where physical actions can cause harm, has already been problematized in earlier safe embodied-AI works [1][2][3]; “safety in VLA” is more accurately a specific instantiation of that broader line, not a fundamentally new problem. It's very inappropriate to have this claim as one of the contributions.

**2. Largely a direct adaptation of existing latent/concept editing recipes.**

The technical core of the proposed method is like: LLM-generated concept stimuli → activation collection → dictionary over concept directions → sparse projection at inference time → coefficient-level gating by an LLM-assigned harmful score, is very close to what has been explored recently for LLM/VLM inference-time safety and for concept-bottleneck / sparse-autoencoder style editing [4][5]. The manuscript does not exploit VLA-specific structure (e.g., action head, temporal rollout, embodiment constraints, or multi-frame perception) and operates only on the convenient fused latent. I don't think it presents enough novelty.

**3. Partially circular / self-serving evaluation.**

The main evidence comes from a safety-augmented Libero variant that the authors themselves construct. That benchmark introduces hazards by injecting explicit, concept-level unsafe instructions written in the same style that the authors use to build their dictionary (LLM-generated, Libero-like stimuli). In other words, the evaluation distribution is made to match the method’s own concept space; a defense that gates “gasoline,” “toxic,” or “child” will obviously look strong on a benchmark where hazards are exactly “put the gasoline can on the hot stove.”

This weakens the claim that the method “unifies” or “generalizes” safety: what we see is good performance under matched concept assumptions. A more convincing evaluation would require, e.g.,  distribution shifts in phrasing and task style, tests on hazards that are not single-token or single-concept, etc.

**4. Safety semantics are LLM-driven and single-concept, but this is not analyzed.**

The entire pipeline depends on two LLM steps: generating stimuli and assigning harmfulness scores to concepts. The paper does not audit the quality or stability of these labels (e.g., across LLMs, prompts, or domains), does not test compositional hazards (two individually safe concepts that become unsafe together), and does not analyze visual-only unsafe inputs. It lacks so many in-depth analyses.

**5. Very important: No real-robot or hardware-in-the-loop validation**

The paper motivates the problem via physical harm and embodied deployment, but all experiments are in simulation / offline VLA benchmarks. Defending attacks on simulation and on real-world are two fundamentally different things.

---

*References:*

[1] Badrobot: Jailbreaking llm-based embodied ai in the physical world. 2024.

[2] Don’t Let Your Robot be Harmful: Responsible Robotic Manipulation via Safety-as-Policy. 2024.

[3] Exploring the Adversarial Vulnerabilities of Vision-Language-Action Models in Robotics. ICCV 2025.

[4] PSA-VLM: Enhancing Vision-Language Model Safety through Progressive Concept-Bottleneck-Driven Alignment. 2024.

[5] Sparse Concept Bottleneck Models: Gumbel Tricks in Contrastive Learning. 2024.

**Questions:**

Please see the weaknesses.

---

### Official Review · Reviewer_SyFU · 2025-10-31

**Soundness:** 1
**Presentation:** 2
**Contribution:** 2
**Rating:** 2
**Confidence:** 4

**Summary:**

This paper proposes SAFE-Dict, a concept-based dictionary learning framework for inference-time safety control in Embdodied systems.

The main idea is to construct a concept dictionary by extracting interpretable latent directions (“concept atoms”) from hidden activations of a pretrained VLA backbone.

At inference time, the hidden state is projected onto this dictionary using ElasticNet decomposition, and the coefficients are used to identify and suppress unsafe concept activations when their cumulative “harmful score” exceeds a threshold.
The paper claims that this framework: (1) is plug-and-play without retraining, (2) generalizes across different VLA architectures, and (3) achieves state-of-the-art defense performance against both explicit unsafe instructions and adversarial jailbreak attacks on benchmarks such as Libero-Harm, BadRobot, RoboPAIR, and IS-Bench.

**Strengths:**

- Addresses a timely and under-explored problem: inference-time safety in embodied AI

- Concept-dictionary-based latent intervention is an interesting and interpretable idea.

- Plug-and-play design without the need for retraining.

- Attempts to model semantic risk structure in latent space.

- Includes preliminary evaluations across multiple safety-relevant embodied benchmarks.

**Weaknesses:**

### Conceptual misunderstanding of VLA systems

The paper demonstrates a fundamental conceptual ambiguity in its definition of Vision–Language–Action (VLA) models.
While the authors repeatedly describe their target as “VLA systems,” the experimental setup and the underlying models, such as `Qwen-VL-72B` and `LLaMA-3.2-Vision`. This suggest that the framework operates on VLM/LLM-driven agents rather than true end-to-end VLA architectures.

A genuine VLA system refers to an integrated perception–language–action pipeline trained on large-scale triplets of (vision, instruction, action) data, in which the model directly maps sensory inputs and linguistic commands to executable motor outputs.
In contrast, VLM-driven agents primarily function as planners or high-level policy generators, which then rely on an external controller or simulator for actuation. Consistent with the setups adopted in this paper (e.g., `BadRobot` and `IS-Bench`), the evaluations concern `jailbreaking embodied LLMs in the physical world` or `VLM-driven embodied agents`, rather than jailbreaking VLA systems.

Consequently, the SAFE-Dict framework, as evaluated on benchmarks such as` BadRobot` and `IS-Bench`, operates on VLM-based embodied agents  rather than true VLAs.
This conceptual misalignment undermines the central claim of the paper, as the demonstrated safety mechanism does not address the end-to-end visuomotor safety problem that the title and abstract imply.

### The experimental section lacks rigorous methodological clarity and internal consistency.

Although the authors report quantitative results on Libero-Harm, BadRobot, RoboPAIR, and IS-Bench, the descriptions of dataset composition, evaluation metrics, and experimental configurations remain substantially under-specified.

The paper claims to have “extended `Libero-10` and `Libero-90` into a safety benchmark” (Section 4.1), yet provides no explicit definition of the new dataset including task categories, risk taxonomy, or data split criteria are absent.

### Limited and biased understanding of safety in embodied intelligence systems

Embodied AI operates in open, dynamic physical environments where safety risks arise not only from malicious or unsafe instructions but also from perception errors, control instability, and unexpected environmental interactions.

The proposed SAFE-Dict framework restricts safety to language-level understanding without addressing the broader system-level safety landscape. By framing safety solely as a problem of malicious instruction detection, the paper significantly reduces its conceptual novelty and practical relevance to real-world embodied systems.

**Questions:**

### Clarifying the scope: VLM-driven agents or true VLA systems?

The problem this paper tackles is undeniably important, and I really appreciate the effort to address safety in embodied AI. That said, I’d encourage the authors to more clearly define whether the systems under consideration are VLM-driven embodied agents or end-to-end Vision–Language–Action (VLA) systems.

From the methodology and experimental desig including the use of `BadRobot` and `IS-Bench`, it seems that the models rely on large vision-language models (e.g., `Qwen-VL`, `LLaMA-Vision`) acting as high-level planners, rather than directly mapping visual observations and language inputs to motor actions. In this sense, the framework appears to focus more on jailbreaking prevention in VLM-driven agents, rather than safety in fully-integrated VLA systems.

In fact, exploring inference-time safety in VLM-based embodied agents is an exciting and timely direction. It may simply be helpful to clarify the scope and terminology throughout the paper, so that the contributions are accurately framed and better appreciated by readers from both communities.

### SAFE-Dict is promising and how does it compare to simpler supervised methods?

The SAFE-Dict framework proposes a conceptually elegant way of assessing safety via latent-space representations and interpretable “concept atoms.” I think this is a compelling direction, especially from the standpoint of model interpretability and potential generalization.

Although SAFE‑Dict is introduced as a multimodal safety framework, from the current experiments, its functionality seems limited to identifying and refusing unsafe language inputs. In principle, this task could also be handled by a simple supervised text‑classification model trained to detect whether a command is safe or compliant.
It would be helpful if the authors could clarify whether SAFE‑Dict is specifically designed to defend against jailbreak attacks and malicious instructions, or if it intends to address broader multimodal safety challenges in embodied systems.

---

### Official Review · Reviewer_2QNW · 2025-11-06

**Soundness:** 3
**Presentation:** 3
**Contribution:** 3
**Rating:** 6
**Confidence:** 5

**Summary:**

This paper tackles safety in Vision-Language-Action models with a plug-and-play inference-time defense. The approach builds a concept dictionary by mining entities from VLA training data, generating concept-specific stimuli, and extracting latent representations. At runtime, they project hidden states onto this dictionary using ElasticNet, compute a harmful score based on unsafe concept activations, and attenuate those activations above a threshold. Evaluations on Libero-Harm, BadRobot, RoboPAIR, and IS-Bench show substantial reductions in attack success while preserving task performance, all without retraining.

**Strengths:**

The biggest win here is practicality. This is genuinely plug-and-play, requiring no retraining of massive robot models, which is huge for real deployments. The interpretability is also excellent—you can actually see which concepts are flagged and why interventions happen, which matters enormously for safety-critical systems. The experimental coverage is solid, spanning multiple benchmarks and threat models with thorough ablations on threshold, attenuation, and sparsity parameters. The writing is clear and well-motivated by the observation that VLA action spaces are bounded, making the set of unsafe concepts tractable.

**Weaknesses:**

My main concern is dictionary completeness. The framework depends entirely on VLM concept mining and LLM-generated stimuli, so safety is limited to known risks. What happens with genuinely novel hazards outside the training distribution? The harmful score assignment also feels ad hoc—using an LLM with anchor examples doesn't guarantee robustness across contexts where a "knife" might be safe or dangerous depending on the situation. The intervention only happens at the final decoder layer, which makes me wonder if unsafe intent is already baked in by that point. Finally, there's limited discussion of failure modes where suppressing concepts might cause task failures or unpredictable behavior.

**Questions:**

How do you handle online adaptation when a deployed robot encounters new hazards not in the original dictionary? Can you speak more to context sensitivity—how does the method distinguish "hand the knife to the chef" from "hand the knife to the child" when the harmful score is concept-fixed? What's the real-world latency overhead on actual robot control loops running at 10-30 Hz? Finally, how well do dictionaries transfer across different VLA architectures—does a dictionary learned on OpenVLA generalize to π0 or RT-2?

---

### Note · Authors · 2025-11-13

I have read and agree with the venue's withdrawal policy on behalf of myself and my co-authors.